# Allelic variation of the *Tas1r3* taste receptor gene affects sweet taste responsiveness and metabolism of glucose in F₁ mouse hybrids

**Vladimir O. Murovets**◉, **Ekaterina A. Lukina**◉, **Egor A. Sozontov**◉, **Julia V. Andreeva**◉, **Raisa P. Khropycheva**◉, **Vasiliy A. Zolotarev**◉ *◉

Pavlov Institute of Physiology, Russian Academy of Sciences, Saint Petersburg, Russia

◉ These authors contributed equally to this work.
* zolotarevva@infran.ru

**Data Availability Statement:** All relevant data are within the paper.

**Funding:** Supported by the Russian Foundation for Basic Research, grant 19-015-00121.

## Abstract

In mammals, inter- and intraspecies differences in consumption of sweeteners largely depend on allelic variation of the *Tas1r3* gene (locus *Sac*) encoding the T1R3 protein, a sweet taste receptor subunit. To assess the influence of *Tas1r3* polymorphisms on feeding behavior and metabolism, we examined the phenotype of F₁ male hybrids obtained from crosses between the following inbred mouse strains: females from 129SvPasCrl (129S2) bearing the recessive *Tas1r3* allele and males from either C57BL/6J (B6), carrying the dominant allele, or the *Tas1r3*-gene knockout strain C57BL/6J-*Tas1r3^{tm1Rfm}* (B6-*Tas1r3*-/-). The hybrids 129S2B6F1 and 129S2B6-*Tas1r3*-/-F1 had identical background genotypes and different sets of *Tas1r3* alleles. The effect of *Tas1r3* hemizygosity was analyzed by comparing the parental strain B6 (*Tas1r3* homozygote) and hemizygous F₁ hybrids B6 × B6-*Tas1r3*-/-. Data showed that, in 129S2B6-*Tas1r3*-/-F1 hybrids, the reduction of glucose tolerance, along with lower consumption of and lower preference for sweeteners during the initial licking responses, is due to expression of the recessive *Tas1r3* allele. Hemizygosity of *Tas1r3* did not influence these behavioral and metabolic traits. However, the loss of the functional *Tas1r3* allele was associated with a small decline in the long-term intake and preference for sweeteners and reduction of plasma insulin and body, liver, and fat mass.

## Introduction

Perception of sweet taste of natural sugars, certain amino acids, and artificial sweeteners acts as a major determinant for strong preference and overconsumption of sweet [1], leading to risks for obesity, type 2 diabetes, nonalcoholic fatty liver disease, and cardiovascular complications [2, 3]. Investigations of the last decade suggest an involvement of the sweet taste receptor in the hormonal regulation of metabolism, in addition to its role in sweet taste sensing for detection of calorie-rich products.

There is good evidence that the sweet taste receptor proteins, along with several signal-transducing molecules, are expressed not only in taste buds on the tongue but also in organs regulating metabolism [4]. Sweet taste receptor subunits T1R2 and T1R3 are coexpressed in

**Competing interests:** The authors have declared that no competing interests exist.

enteroendocrine cells of the small intestine [5, 6]. Their activation by natural sugars and artificial sweeteners leads to secretion of gut hormones such as glucagon-like peptide-1 (GLP-1), GLP-2, and the gastric inhibitory peptide (GIP), acting as incretins to enhance insulin secretion [5, 7]. In the intestine, T1R3 upregulates through GLP-1 and GIP expression of sodium-dependent glucose transporter isoform 1 (SGLT1) important for the provision of glucose to the body and avoidance of intestinal malabsorption [6]. Coupling of T1R3 with T1R1 produces a heterodimer that serves as an intestinal L-amino acid sensor modulating cholecystokinin release [8].

The T1R3 protein is detected in mouse pancreatic β-cells, where its expression level is greater than that of T1R2, suggesting that most likely T1R3 is present in homodimeric form or in more sensitive to glucose heteromers when coupled with a calcium-sensing receptor [9–11]. In pancreatic β-cells, stimulation of the sweet taste receptor elicits insulin release by elevating intracellular $Ca^{2+}$ and/or cAMP [12] synergetic to the major universal mechanism of metabolic detection of glucose, which includes the glucose transporter GLUT2, glucokinase, and the ATP-sensitive $K^+$ ($K_{ATP}$) channel [13]. In β-cells, T1R3 can be also dimerized with the T1R1 subunit, forming the receptor detecting amino acids, which stimulates insulin synthesis and reduces cell apoptosis, affecting the mechanistic target of rapamycin complex 1 (mTORC1) through activation of phospholipase C and extracellular signal-regulated protein kinase 1 (ERK1) and ERK2 [14]. Importantly, the knockout of T1R3 in mice resulted in substantial impairment of metabolism and cell proliferation. The *Tas1r3-/-* deletion decreased glucose tolerance [15, 16] and increased insulin resistance [17]. T1R3-knockout animals are resistant to sucrose-induced obesity and have smaller fat depots when fed a high-sucrose diet [15]. Deletion of *Tas1r3-/-* reduces activation of mTORC1 [14] and decreases density of pancreatic islets and expression of caspase 3 [18].

The considerable variation in perception and consumption of sweet taste compounds within mammalian species is mainly genetically determined. The analysis of heritable differences in sweet taste responses revealed that inbred strains of mice with higher preference for a large number of chemically diverse sweeteners had the so-called taster allele and that strains with relatively lower sensitivity had the nontaster allele [19, 20]. These differences are associated mainly with the autosomal locus named *Sac* (saccharin preference). The dominant allele of this locus (*Sac^b^*) found in the C57BL/6 strain is associated with higher saccharin preference, while the recessive allele (*Sac^d^*) is linked to lower saccharin preference [21, 22]. Positional cloning of the *Sac* locus in the subtelomeric region of mouse chromosome 4 has shown its correspondence to the *Tas1r3* gene, which encodes the T1R3 receptor protein [19].

Loss of T1R genes or their pseudogenization in several vertebrate species, as well as artificial gene deletion, results in sharp decreases in perception and intake of sweeteners [23]. One analysis found that about 78% of differences among mouse strains in preference for saccharin are explained by amino acid sequence variation in the T1R3 protein. Allelic variants of the *Tas1r3* gene correspond mainly to three nonsynonymous single nucleotide polymorphisms (SNPs) which do not act by blocking gene expression, changing alternative splicing, or interfering with protein translation in taste tissue. Among the polymorphisms, T179C, which causes a substitution of isoleucine to threonine at position 60 in the extracellular N domain of the T1R3 protein, influence the ability of the protein to form dimers or bind sweeteners [24]. This substitution reduces *in vitro* binding of T1R3 to caloric or noncaloric sweeteners, increasing, for instance, effective dose for sucrose up to 1000% [25]. Synonymous or nonsynonymous SNPs were also found in human *TAS1R* genes, as well as haplotypes of separate populations [26, 27]. These polymorphisms may be associated with sweet taste sensitivity and carbohydrate consumption [28, 29].

Evidence for interactions of *Tas1r3* polymorphisms with preference and consumption of nutritive or nonnutritive sweeteners was obtained using a congenic 129.B6-*Tas1r3* strain [30]

or sequencing *Tas1r3*-containing fragments of DNA in strains with marked differences in sweetener preference [24]. The genetic architecture of postoral detection of glucose and artificial sweeteners is more complicated than inborn determinants affecting oral taste responses; thus, contributions of *Tas1r3* polymorphisms likely are masked by background genotype variability. For example, in $F_2$ crosses between C57BL/6 and 129P3 strains, variation in sweetener intake depended less on *Tas1r3* genotype (10–35%) than did variation in preference (64–96%) [31, 32].

The goal of the current study was to define *in vivo* influences of *Tas1r3/Sac* allelic variation, against the mixed 129S2B6F1 genetic background, on carbohydrate metabolism and body composition, which to our knowledge has not yet been assessed. For this purpose, we undertook the investigation using mouse $F_1$ hybrids between 129SvPasCrl (129S2) and C57BL/6J (B6) inbred strains, or between 129S2 and *Tas1r3*-knockout C57BL/6J-*Tas1r3*<sup>tm1Rfm</sup> (B6-*Tas1r3*-/-) strains. The inbred B6 strain is homozygous for the *Sac<sup>b</sup>* "taster" (T) allele associated with high avidity for sweeteners [23], while the 129S was among the first strains described as sweet "nontasters" (NT) [22, 33] carrying the recessive *Sac<sup>d</sup>* allele with nucleotide sequence identical to other nontaster strains [34].

Studies have shown that plasma glucose level was considerably higher in the B6 strain (phenome.jax.org/projects/Eumorphia3/strains) than in 129 [35, 36]. Mice from the B6 strain show mild glucose intolerance and impaired insulin secretion despite normal insulin sensitivity [37]. In contrast, the 129 substrains (phenome.jax.org/projects/Eumorphia3/strains) exhibited higher glucose tolerance [35]. Likewise, the B6 mice were more insulin resistant during the insulin tolerance test, whereas 129SVE mice were relatively insulin sensitive [35]. Unexpectedly, we found no reliable data on glucose tolerance for the 129P3/J strain, which is why we selected 129S2 mice as the parental strain. The *Tas1r3*-knockout (B6-*Tas1r3*-/-) strain lacking the entire T1R3 coding region does not demonstrate behavioral or nerve responses to sweet substances [11, 22, 38]. Furthermore, the B6-*Tas1r3*-/- mice have decreased glucose tolerance compared to the parental strain [15, 16].

By mating females from the 129S2 strain with males from B6 or *B6-Tas1r3*-/- strain, we obtained $F_1$ hybrids with identical background genotypes but carrying different *Tas1r3* alleles: either both recessive NT and dominant T alleles, *Sac<sup>b</sup>* and *Sac<sup>d</sup>* (129S2B6F1), or the single recessive NT allele *Sac<sup>d</sup>* (129S2B6-*Tas1r3*-/-F1). Taking into account dominance of the *Sac<sup>b</sup>* allele [21], we hypothesized differences between hybrids in oral and postoral detection of sweet compounds leading to variations in carbohydrate metabolism. Additionally, we examined the effect of *Sac* hemizygosity, comparing phenotypes of the B6 parental strain (*Sac<sup>b</sup>* homozygote) with the crosses B6 × B6-*Tas1r3*-/-. Earlier it was shown that T1R3-mediated increase in insulin secretion from pancreatic islets required an optimal concentration of glucose in the media matching the postprandial level in blood [9, 39]. Therefore, in our experiments we used nonfasted animals having free access to the normocaloric lab chow during the light period.

## Materials and methods

All experimental procedures have been approved by the Animal Care and Use Committee at the Pavlov Institute of Physiology (Animal Welfare Assurance #A5952-01). The protocols were designed in accordance with the National Institutes of Health Guidelines for the Care and Use of Laboratory Animals.

### Animals, housing, and diet

Subjects were adult mice of inbred strains and their intercrosses. C57BL/6J (B6) and 129SvPasCrl (129S2) mice were derived from the parental stocks obtained from the Jackson

Laboratory (Bar Harbor, ME, USA) and Charles River Laboratory (Sulzfeld, Germany), respectively. The C57BL/6J-*Tas1r3*<sup>tm1Rfm</sup> *Tas1r3*-knockout mice were offspring of the breeding group kindly provided by Dr. R. F. Margolskee (Monell Chemical Senses Center, Philadelphia, PA, USA). Insulin and triglyceride measurements were made in hybrids between C57BL/6ByJ (B6By) and 129P3/J (129P) mice.

Experiments were performed with male mice at 3–7 months of age maintained at the vivarium of the Pavlov Institute. Animals were housed individually during taste tests, or otherwise in groups of 4–5 in standard polycarbonate cages in a temperature- and humidity-controlled room on a 12/12-h light/dark cycle. Laboratory chow (PK-120, MEST Ltd., Moscow, Russia) containing 67% carbohydrate, 5% lipids, and 19% protein, with an energy value of 13000 kJ/kg, and tap water were available *ad libitum*, unless otherwise specified.

## Taste tests

Behavioral taste responses to sucrose or noncaloric sweeteners were examined in separate groups of mice using the brief-access licking test (Group 1: 129S2B6F1, n = 6; 129S2B6-*Tas1r3*-/-F1, n = 5; B6, n = 19; B6-*Tas1r3*+/-, n = 29) or the long-term two-bottle choice test (Group 2: 129S2B6F1, n = 11; 129S2B6-*Tas1 r3*-/-F1, n = 13; B6, n = 22; B6-*Tas1r3*+/-, n = 23). The brief-access licking test (BALT) was conducted in the middle of the light period (1–3 p.m.) using procedures described by J. Glendinning et al. [40], with minor modifications. During the experimental session, an animal was exposed in a Davis MS-160 gustometer (DiLog Instruments, Tallahassee, FL, USA), in which a sipper tube containing test solution was presented for 5 s, with 20-s interpresentation interval. Animals first had two training sessions with water at 1-day intervals. On training day 1, the animal could drink water freely for 30 min from a single spout. On training day 2, the mouse had 24 trials with water, each lasting for 5 seconds. After each training day, animals had free access to water for 60 min in their home cages. To motivate licking from the sipper tube, animals were deprived of water for 22–23 h before training. To encourage drinking of sweet solutions, we limited the mice to 1.5 mL of water for 22–23 h prior the test session.

In the BALT, animals had access to solutions of sucrose (0.5–4%; Vecton Ltd., St. Petersburg, Russia), sodium saccharinate (0.2–60 mmol/L; Tiaujin Chaugjie Chem. Co., Ltd., China), sucralose or potassium acesulfame (both 0.3–10 mmol/L; Sigma Aldrich Corp., St. Louis, MO, USA) dissolved in deionized water and presented at room temperature. Sweet taste substances were tested at day intervals. The sequence of testing of substances was the same for all animals.

In the 48-h two-bottle choice test (2-BT), individually housed mice from Group 2 had unlimited access to two tubes, one containing solution of sucrose (1–16%) or saccharin (0.2–60 mmol/L), and the other filled with deionized water [41]. Each concentration of solution was given for 48 h. Every 24 h tubes were weighed to the nearest 0.02 g and their position in the cage was changed. Average daily intakes were calculated for each concentration. Preference score was determined as the ratio (in percent) of daily intake of test solution to daily intake of fluid (solution + water). Solutions of sucrose and saccharin were tested in separate groups in ascending order of concentrations.

## Glucose and insulin tolerance tests

Glucose and insulin tolerance tests were performed in the middle of the light period in a separate group of nonfasted awake male mice, including 129S2B6F1 (n = 21), 129S2B6-*Tas1r3*-/-F1 (n = 22), B6 (n = 42), and B6-*Tas1r3*+/- (n = 36). These animals were kept in their home cages. In the glucose tolerance test (GTT), matching specific recommendations of the National

Institutes of Health Mouse Metabolic Phenotyping Center [42], 2 g/kg glucose (Sigma Aldrich Corp.) was administered in aqueous solution either by intragastric gavage (IG) or intraperitoneally (IP). In the insulin tolerance test (ITT), animals were injected IP with 2 U/kg insulin (insulin aspart, Novo Nordisk A/S, Bagsvaerd, Denmark). Blood was sampled by tail cut, and two measurements of glucose concentration were made at 0, 10, 15, 30, 60, 90, and 120 minutes after injection of glucose or at 0, 15, 60, and 120 minutes after administration of insulin, using the OneTouch® UltraEasy® glucometer (LifeScan Europe Ltd., Division of the Cilag Gmbh Int., Zug, Switzerland) and OneTouch® Ultra® test strips (LifeScan Scotland Ltd., Inverness, Great Britain).

### Biochemical assay and morphometry

Blood sampling for estimation of hormone and metabolite levels was done immediately after sedation with gas mixture of $CO_2$ and $O_2$ (50/50 volume %) in a separate group of nonfasted awake male mice, including 129PB6ByF1 (n = 18), 129PB6-*Tas1r3*-/-F1 (n = 17), B6 (n = 19), and B6-*Tas1r3*+/- (n = 21). These animals were kept in their home cages. Blood was withdrawn from the orbital sinus into tubes containing EDTA and then centrifuged for plasma separation. Concentrations of triglycerides and glycerol were assessed in fresh plasma using the TR0100 assay kit (Sigma Aldrich Corp.); other samples were kept at -30˚C. Plasma insulin level was measured with Elisa kit EZRMI-13K (Millipore Corp., Burlington, MA, USA).

Body composition was assessed in a group of naïve mice, as well as mice used in the BALT 3 weeks earlier. After euthanasia by decapitation, liver and fat were removed and weighed to the nearest 0.001 g. The anterior subcutaneous (interscapular), posterior subcutaneous (dorsolumbar, inguinal, and gluteal), visceral perirenal visceral mesenteric, and retroperitoneal visceral epididymal (gonadal) bilateral depots were excised.

### Data analysis

Plasma glucose area under the curve (AUC) was calculated using the trapezoidal rule. Data from behavioral taste tests and glucose levels in GTT and ITT were compared using two-way ANOVA (Statistica, version 7.0). Concentration (for taste tests) and time (for GTT or ITT) were considered as within-subject factors, and genotype as a between-subject factor. Post hoc paired comparisons were made using Fisher's least significant difference (LSD) test. Differences between AUCs or water consumption in training sessions for the BALT were determined with one-way ANOVA. Other comparisons were made with unpaired two-tailed Student's *t*-test. All data are presented as mean ± SEM; *p*-values < 0.05 are considered significant. ANOVA results are listed in Table 2.

## Results

### Licking response

In the training sessions of the BALT, we found no differences in water intake between 129S2B6F1 and 129S2B6-*Tas1r3*-/-F1 hybrids. Interestingly, mice from the 129S2 parental strain showed water consumption similar to hybrid groups. The B6-*Tas1r3*+/- hemizygotes drank less water during the first training session than animals from the B6 strain, but both genotypes showed a similar lick rate the next day (Table 1). These results justify the following comparison of raw lick rates between 129S2B6F1 and 129S2B6-*Tas1r3*-/-F1 or between B6 and B6-*Tas1r3* +/-mice without converting licking number per trial into a standardized lick ratio [43].

Mice from all groups exhibited robust concentration-dependent increases in licking rate for all four tested sweet compounds (ANOVA results are shown in the Table 2). Compared with

**Table 1. Water consumption and licking rates of mice with different *Tas1r3/Sac* genotypes during training sessions in the gustometer.**

| Genotype | *n* | First training session | | Second training session | |
|---|---|---|---|---|---|
| | | Water intake (g) | Licks/30 min | Water intake (g) | Licks/trial |
| **129S2B6F1** | 11 | 0.86±0.07 | 475±37 | 0.99±0.15 | 28.7±2.06 |
| **129S2B6-*Tas1r3*-/-F1** | 10 | 0.82±0.08 | 477±59 | 0.87±0.05 | 25.8±1.6 |
| **129S2** | 5 | 0.79±0.07 | 459±41 | 0.78±0.12 | 28.2±3.3 |
| **B6** | 19 | 0.82±0.08 [a] | 536±53 [b] | 0.89±0.04 | 23.5±1.0 |
| **B6-*Tas1r3*+/-** | 28 | 0.64±0.05 [a] | 403±33 [b] | 0.90±0.06 | 25.1±1.1 |

Differences between genotypes were assessed with One-way ANOVA followed by Fisher's LSD post hoc test. Values with the same letters are significantly different at $p<0.05$.

**Table 2. ANOVA results for Figs 1–4.**

| *Fig #* | ANOVA Effect | *F*(DFn, DFd) | P Value | *Fig #* | ANOVA Effect | *F*(DFn, DFd) | P Value |
|---|---|---|---|---|---|---|---|
| Fig 1A | Strain | $F(1, 18) = 7.24$ | P < 0.015 | Fig 2F | Strain | $F(1, 22) = 6.78$ | P < 0.02 |
| | Concentration | $F(4, 72) = 36.80$ | P < 0.00001 | | Concentration | $F(4, 88) = 109.40$ | P < 0.00001 |
| | Interactions | $F(4, 72) = 12.20$ | P < 0.00001 | | Interactions | $F(4, 88) = 1.48$ | P > 0.21, N.S. |
| Fig 1B | Strain | $F(1, 8) = 13.18$ | P < 0.01 | Fig 2G | Strain | $F(1, 19) = 5.56$ | P < 0.03 |
| | Concentration | $F(6, 48) = 7.90$ | P < 0.00001 | | Concentration | $F(5, 95) = 83.58$ | P < 0.00001 |
| | Interactions | $F(4, 72) = 9.78$ | P < 0.00001 | | Interactions | $F(5, 95) = 1.95$ | P > 0.09 |
| Fig 1C | Strain | $F(1,7) = 9.92$ | P < 0.02 | Fig 2H | Strain | $F(1, 19) = 17.47$ | P < 0.001 |
| | Concentration | $F(4, 28) = 22.16$ | P < 0.00001 | | Concentration | $F(5, 95) = 121.83$ | P < 0.00001 |
| | Interactions | $F(4, 28) = 10.94$ | P < 0.00005 | | Interactions | $F(5, 95) = 1.82$ | P > 0.11, N.S. |
| Fig 1D | Strain | $F(1, 9) = 17.84$ | P < 0.003 | Fig 3 | Strain | $F(4, 70) = 45.85$ | P < 0.0001 |
| | Concentration | $F(4, 36) = 12.34$ | P < 0.00001 | | Time | $F(7, 490) = 172.39$ | P < 0.0001 |
| | Interactions | $F(4, 36) = 12.85$ | P < 0.00001 | | Interactions | $F(28, 490) = 11.40$ | P < 0.001 |
| Fig 1E | Strain | $F(1, 46) = 0.11$ | P > 0.74, N.S. | | AUC | $F(4, 70) = 42.23$ | P < 0.0001 |
| | Concentration | $F(4, 184) = 128.76$ | P < 0.00001 | Fig 4A | Strain | $F(1, 16) = 11.42$ | P < 0.005 |
| | Interactions | $F(4, 184) = 0.82$ | P > 0.52, N.S. | | Time | $F(7, 112) = 47.08$ | P < 0.00001 |
| Fig 1F | Strain | $F(1,28) = 2.00$ | P > 0.16, N.S. | | Interactions | $F(7, 112) = 1.70$ | P > 0.11, N.S. |
| | Concentration | $F(6, 168) = 31.74$ | P < 0.00001 | | AUC | $F(1, 16) = 14.11$ | P < 0.002 |
| | Interactions | $F(6, 168) = 1.23$ | P > 0.29 N.S. | Fig 4B | Strain | $F(1, 18) = 3.48$ | P > 0.08, N.S. |
| Fig 1G | Strain | $F(1,33) = 0.67$ | P > 0.41, N.S. | | Time | $F(7, 126) = 36.02$ | P < 0.00001 |
| | Concentration | $F(4, 132) = 122.67$ | P < 0.00001 | | Interactions | $F(7, 126) = 4.70$ | P < 0.0001 |
| | Interactions | $F(4, 132) = 1.86$ | P > 0.12, N.S. | | AUC | $F(1, 18) = 5.52$ | P < 0.03 |
| Fig 2A | Strain | $F(1, 13) = 56.74$ | P < 0.0001 | Fig 4C | Strain | $F(1, 26) = 0.85$ | P > 0.36, N.S. |
| | Concentration | $F(4, 52) = 124.67$ | P < 0.00001 | | Time | $F(7, 182) = 105.50$ | P < 0.00001 |
| | Interactions | $F(4, 52) = 13.60$ | P < 0.00001 | | Interactions | $F(7, 182) = 0.72$ | P > 0.65, N.S. |
| Fig 2B | Strain | $F(1, 13) = 60.27$ | P < 0.0001 | | AUC | $F(1, 27) = 1.17$ | P > 0.28, N.S. |
| | Concentration | $F(4, 52) = 110.68$ | P < 0.00001 | Fig 4D | Strain | $F(1, 30) = 6.02$ | P < 0.02 |
| | Interactions | $F(4, 52) = 13.45$ | P < 0.00001 | | Time | $F(7, 210) = 60.81$ | P < 0.00001 |
| Fig 2C | Strain | $F(1, 19) = 31.16$ | P < 0.0005 | | Interactions | $F(7, 210) = 2.69$ | P < 0.011 |
| | Concentration | $F(5, 45) = 12.76$ | P < 0.00001 | | AUC | $F(1, 30) = 5.88$ | P < 0.025 |
| | Interactions | $F(5, 45) = 9.61$ | P < 0.00001 | Fig 4E | Strain | $F(1, 12) = 0.16$ | P > 0.69, N.S. |
| Fig 2D | Strain | $F(1, 19) = 10.41$ | P < 0.01 | | Time | $F(3, 36) = 306.15$ | P < 0.00001 |
| | Concentration | $F(5, 45) = 7.98$ | P < 0.00001 | | Interactions | $F(3, 36) = 0.78$ | P > 0.51, N.S. |
| | Interactions | $F(5, 45) = 5.77$ | P < 0.0005 | Fig 4F | Strain | $F(1, 15) = 0.22$ | P > 0.64, N.S. |
| Fig 2E | Strain | $F(1, 22) = 13.7$ | P < 0.0015 | | Time | $F(3, 45) = 156.15$ | P < 0.00001 |
| | Concentration | $F(4, 88) = 108.90$ | P < 0.00001 | | Interactions | $F(3, 45) = 0.87$ | P > 0.46, N.S. |
| | Interactions | $F(4, 88) = 2.57$ | P < 0.05 | | | | |

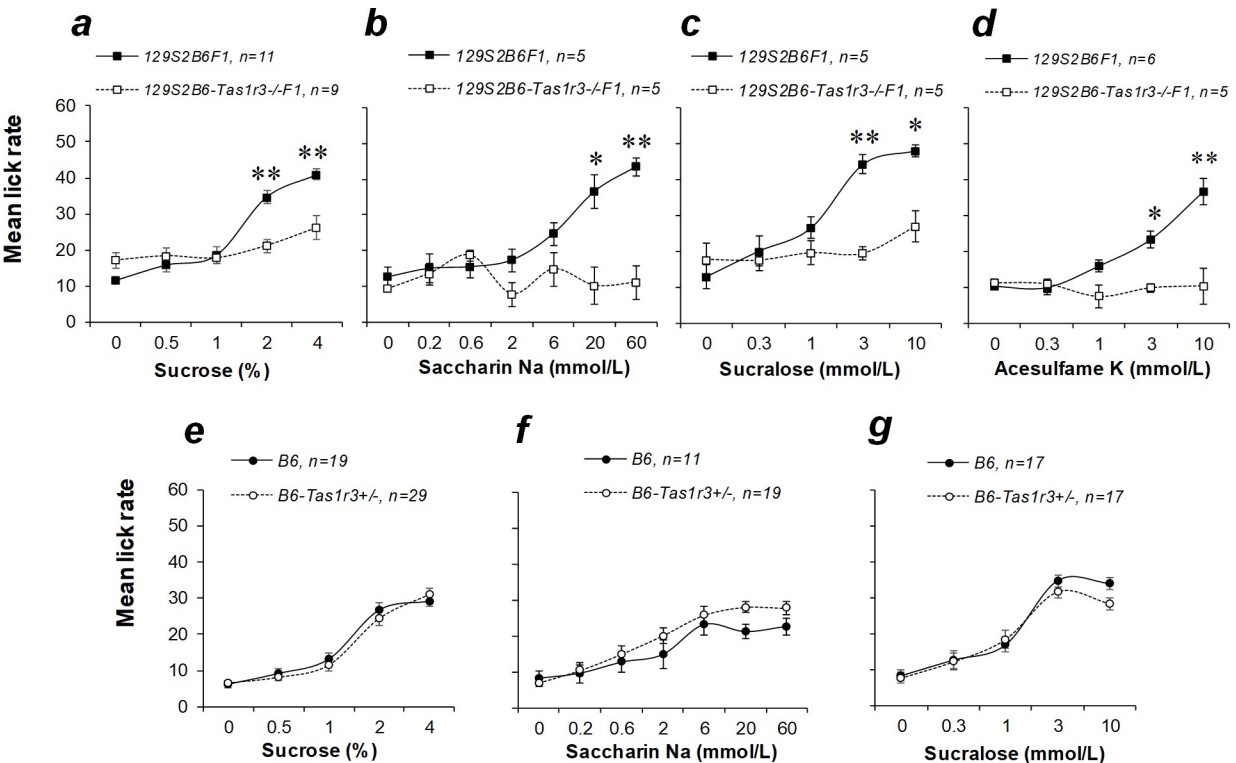

**Fig 1.** Initial licking responses to solutions with a range of concentrations of sucrose and artificial sweeteners in the brief-access testing of 4- to 7-month-old male mice with different *Tas1r3* genotypes: 129S2B6F1 and 129S2B6-*Tas1r3*-/-F1 (*a–d*), or B6 and B6-*Tas1r3*+/- (*e–g*). Post hoc comparisons with Fisher's LSD test: *$p < 0.05$, **$p < 0.01$.

the 129S2B6-*Tas1r3*-/-F1 hybrids, mice from 129S2B6F1 group showed enhanced lick responsiveness to certain concentrations (Fig 1A–1D). The 129S2B6F1 hybrids licked 2–4% sucrose solutions more vigorously than did 129S2B6-*Tas1r3*-/-F1 mice (Fig 1A; $p < 0.01$, Fisher's LSD test). When animals were offered solutions of saccharin, sucralose, or acesulfame, 129S2B6F1 mice displayed more active licking of higher concentrations than did 129S2B6-*Tas1r3*-/-F1 mice (Fig 1B–1D). An influence of *Tas1r3* hemizygosity was assessed by matching reactions of the B6 strain (*Tas1r3* homozygote) and B6-T*as1r3*+/- mice, carrying a single *Tas1r3* allele. We did not detect significant signs of haplo-insufficiency at any tested concentration of either caloric (sucrose) or noncaloric (saccharin and sucralose) sweeteners (Fig 1E–1G).

## Long-term two-bottle sweetener preference

The 2-BT was used to examine preference and intake of sucrose and saccharin. For both sweeteners, the 129S2B6F1 hybrids demonstrated overall higher intakes and preference scores than did 129S2B6-*Tas1r3*-/-F1 mice (Fig 2A–2D; Table 2). Paired post hoc comparisons confirmed that 129S2B6F1 mice consumed larger amounts of intermediate and high concentrations of sucrose (4–8%) and showed higher preference for sucrose solutions at a relatively lower (1–4%) concentration range (Fig 2A and 2B; $p < 0.05$). 129S2B6-*Tas1r3*-/-F1 hybrids sharply increased consumption of sucrose when they were given 8–16% solutions ($p < 0.01$) and did not differ from 129S2B6F1 mice in preference for 8–16% solutions (Fig 2A and 2B). The 129S2B6F1 hybrids preferred and consumed saccharin more vigorously than did 129S2B6-*Tas1r3*-/-F1 at higher tested concentrations (6–60 mM; Fig 2C and 2D). Assessment of the

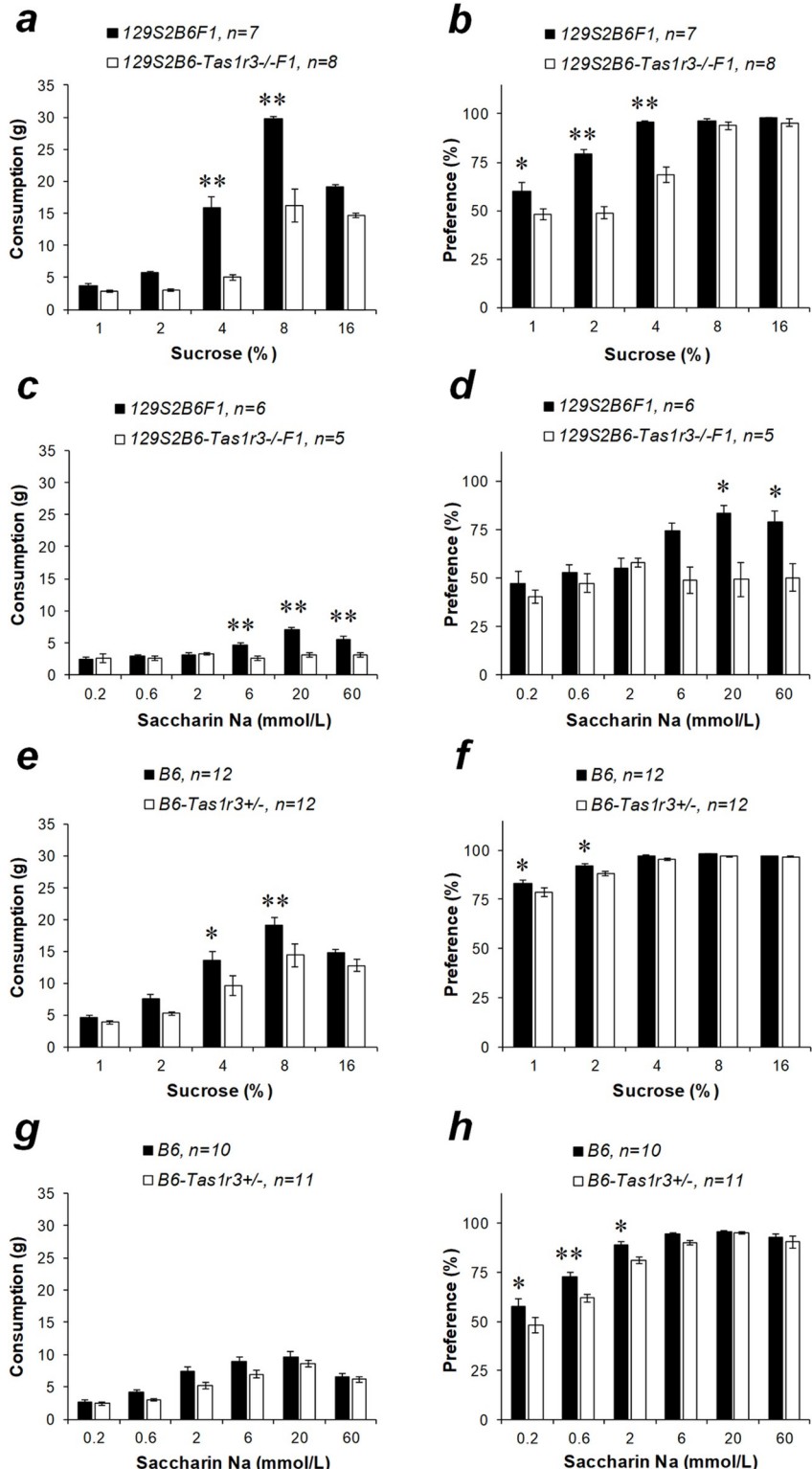

**Fig 2.** Sucrose and noncaloric sweetener intakes and preferences in the two-bottle preference test in 2- to 7-month-old male mice with different *Tas1r3* genotypes: 129S2B6F1 and 129S2B6-*Tas1r3*-/-F1 (*a–d*), or B6 and B6-*Tas1r3*+/- (*e–h*). Post hoc comparisons with Fisher's LSD test: *$p < 0.05$, **$p < 0.01$.

effect of the *Tas1r3* hemizygosity revealed that the loss of a *Tas1r3* allele led to reduced intake of 4–8% sucrose solution and slightly reduced preference score for the lowest tested concentrations (1–2%; Fig 2E and 2F). Additionally, *Tas1r3* hemizygosity was associated with a decrease in preference for lower concentrations of saccharin in the range of 0.2–2 mmol/L but did not affect saccharin consumption (Fig 2G and 2H).

Together, these findings indicate that initial lick responses to nutritive and nonnutritive sweeteners depend mainly on polymorphisms of the *Tas1r3* gene and that the effect of haploinsufficiency of *Tas1r3* is negligible. Presence of the dominant *Tas1r3* allele confers higher lick responsiveness to lower concentrations of sucrose and higher concentrations of saccharin, sucralose, and acesulfame. The preference for sucrose in the 2-BT is also affected by the *Tas1r3* polymorphisms; however, between-genotype differences in long-term consumption of sucrose may partially depend on *Tas1r3* hemizygosity.

## The relationship between *Sac* genotype and glucose and insulin tolerance

Initial analysis of IP glucose tolerance was performed in parental strains: 129P, 129S2, B6, B6By, and B6-*Tas1r3*-/-. In the nonfasting state, parental strains differed in baseline glucose level, which was higher in the B6 strain, including the knockout substrain, than in 129 (Fig 3; p<0.05). However, there was significant overlap between responses of B6 and B6By, as well as between 129P and 129S2. The IP glucose tolerance was substantially reduced in the B6 strain compared to 129 (p<0.005), but no significant differences were found within strains excluding B6-*Tas1r3*-/-. The *Tas1r3* gene deficiency markedly worsens glucose tolerance (Fig 3) compared both to B6 and to B6By (p<0.01).

Comparison of Figs 3 and 4A demonstrates that hybrid groups had intermediate glucose tolerance between parental strains. After both IP and IG glucose load, nonfasted 129S2B6F1 hybrids exhibited higher glucose tolerance than did 129S2B6-*Tas1r3*-/-F1 (Fig 4A and 4B). Interestingly, in B6 and B6-*Tas1r3*+/- mice, plasma glucose concentration was nearly identical after IP injection; however, after IG administration, glucose levels in B6-*Tas1r3*+/- were significantly reduced (Fig 4C and 4D). IP injection of insulin elicited a rapid (<15 min) fall of plasma glucose concentration, which barely recovered during the subsequent 2 h. We found no significant differences in tolerance to insulin between 129S2B6F1 and 129S2B6-*Tas1r3*-/-F1 hybrids or between B6 and B6-*Tas1r3*+/- mice (Fig 4E and 4F).

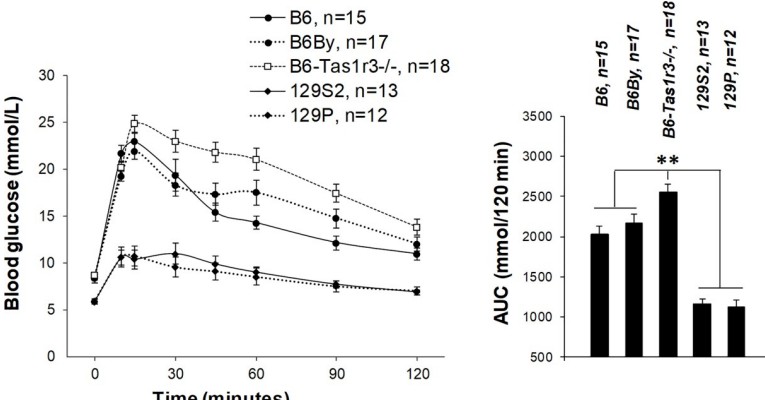

**Fig 3. Glucose tolerance test in 3- to 5-month-old nonfasted male mice of parental strains: B6, B6By, B6-Tas1r3-/-, 129S2, and 129P.** Plasma glucose concentrations and glucose area under the curve (AUC) were determined after intraperitoneal (IP) administration of 2 g/kg glucose. Post hoc comparisons with Fisher's LSD test: **p<0.01.

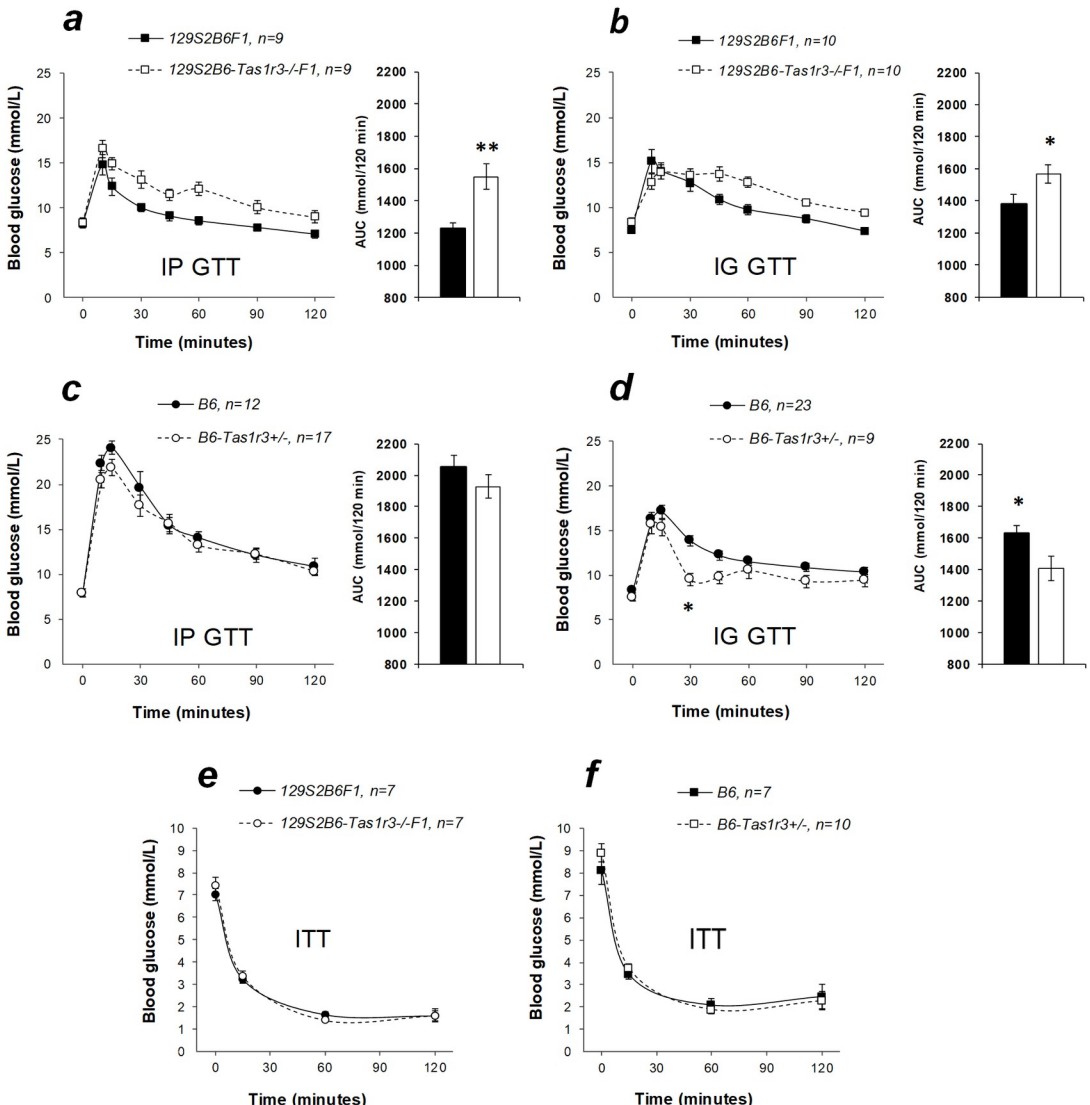

**Fig 4.** Glucose tolerance test (IP and IG GTT; *a–d*) in 3–4-month-old nonfasted male mice and insulin tolerance test (ITT; *e, f*) in 5–6-month-old nonfasted male mice with different *Tas1r3* genotypes: 129S2B6F1 and 129S2B6-*Tas1r3*-/-F1, or B6 and B6-*Tas1r3*+/-. Plasma glucose concentrations and glucose area under the curve (AUC) were determined after the intraperitoneal (IP) or intragastric (IG) glucose administration of 2 g/kg glucose. Post hoc comparisons with Fisher's LSD test: $^*p < 0.05$, $^{**}p < 0.01$.

## Baseline plasma glucose, insulin, triglyceride, and glycerol levels and body composition

Baseline glucose and body composition were assessed in the B6 strain and hybrids 129S2B6F1, 129S2B6-*Tas1r3*-/-F1, and B6-*Tas1r3*+/- in the nonfasting state in the middle of the light period (Fig 5). Concentration of plasma glucose was not affected by *Tas1r3* polymorphisms or hemizygosity for the *Tas1r3* allele (Fig 5A and 5E). We found no differences in body, fat, or liver mass between 129S2B6F1 and 129S2B6-*Tas1r3*-/-F1 mice (Fig 4B–4D). However, the lack of a single functional *Tas1r3* allele in B6-*Tas1r3*+/- was associated with significant increase of the body, fat, and liver mass (Fig 4F–4H).

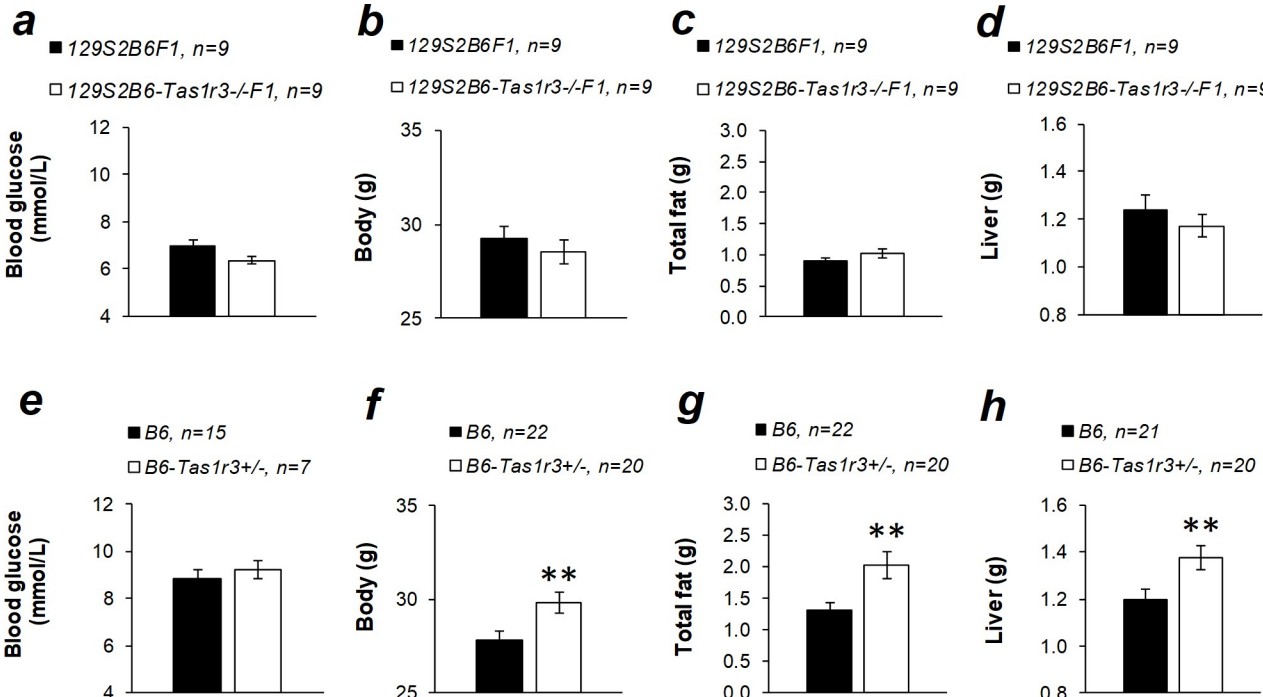

**Fig 5. Plasma glucose concentration and body, fat, and liver mass in nonfasted inbred and hybrid 5- to 6-month-old nonfasted male mice with different *Tas1r3* genotypes: 129S2B6F1 and 129S2B6-*Tas1r3*-/-F1, or B6 and B6-*Tas1r3*+/-.** ${}^{**}p{<}0.01$, Student's *t*-test.

Insulin and metabolite concentrations were measured in plasma of the B6 strain and hybrids 129PB6ByF1, 129PB6-*Tas1r3*-/-F1, and B6-*Tas1r3*+/- in the nonfasting state in the middle of the light period. 129PB6ByF1 hybrids exhibited substantially higher baseline insulin levels compared with 129PB6-*Tas1r3*-/-F1 (Fig 6A). A marked reduction of plasma insulin was found in the B6-*Tas1r3*+/- hemizygotes, unlike B6 homozygotes (Fig 6D). There was no significant difference between all compared genotypes in plasma triglyceride concentration (Fig 6B and 6E). In contrast, the nonfasted plasma level of glycerol depended on the *Tas1r3* genotype: 129PB6-*Tas1r3*-/-F1 hybrids demonstrated higher concentration of glycerol compared with 129PB6ByF1 mice (Fig 6C). Hemizygous B6-*Tas1r3*+/- mice and *Tas1r3*-homozygous B6 mice showed similar plasma glycerol levels (Fig 6F).

## Discussion

In this study, we confirmed that our experimental model reproduces polymorphic influences of the mouse *Tas1r3* genotype on the appetitive responses to sweeteners described in the earlier investigations with 129.B6-*Tas1r3* congenic mice [30]. Using the F$_1$ hybrids between C57BL/6J and 129SvPasCrl strains, which had identical background genotypes but carried different *Tas1r3* alleles, we demonstrated that the presence of the dominant T allele in 129S2B6F1 hybrids is associated with the stronger preference for sweet solutions compared with 129S2B6-*Tas1r3*-/-F1 mice expressing only the recessive NT allele. The effect of the *Tas1r3* genotype depended on concentrations of sweet taste compound that were specific to each sweetener. In the BALT, lick rate for sucrose was influenced by the *Tas1r3* genotype when lower (2–4%) concentrations were tested (Fig 1A). As shown in our previous study in crosses between C57BL/6ByJ and 129P3/J strains, responses to higher concentrations of sucrose (8–32%) were not linked to the *Tas1r3* polymorphisms [44]. At the same time, when animals were exposed to

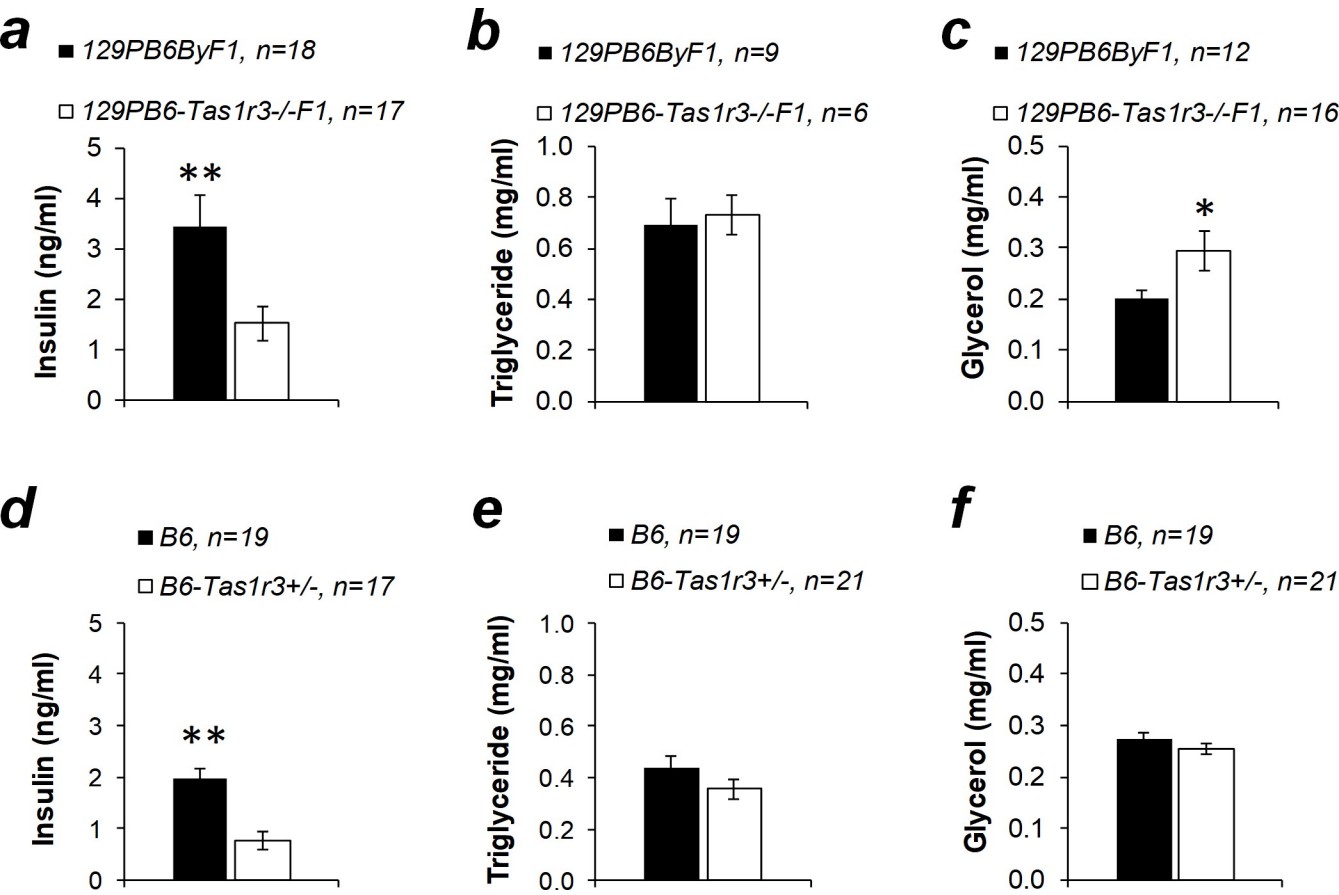

**Fig 6. Plasma insulin, triglyceride, and glycerol concentrations in 4- to 7-month-old nonfasted inbred and hybrid male mice with different *Tas1r3* genotypes: 129PB6ByF1 and 129PB6-*Tas1r3*-/-F1, or B6 and B6-*Tas1r3*+/-.** $^*p < 0.05$, $^{**}p < 0.01$, Student's *t*-test.

nonmetabolized sweeteners (saccharin, sucralose, and acesulfame), the impact of *Tas1r3* polymorphisms tended to be strongest at the higher concentration range (Fig 1B–1D).

The *Tas1r3* gene polymorphisms also affected long-term intake and preference scores for sweeteners in the 48-h 2-BT. 129S2B6F1 hybrids carrying the dominant *Tas1r3* allele demonstrated a marked increase in consumption of low to intermediate concentrations (4–8%) and preference for lower (1–4%) concentrations of sucrose, compared with hybrids having the recessive allele. However, allelic variation of the *Tas1r3* gene did not influence the preference for higher concentrations (8–16%) of sucrose (Fig 2A and 2B). The 129S2B6F1 mice consumed larger amounts and showed greater preference scores for higher concentrations of saccharin compared to 129S2B6-*Tas1r3*-/-F1 (Fig 2C and 2D).

These data are generally consistent with results of earlier studies of responsiveness to different concentrations of sweeteners, determined by *Tas1r3* allelic variations, which were performed with inbred mouse strains bearing T or NT *Tas1r3* alleles [40] or the 129.B6-*Tas1r3* segregating congenic strain [30]. The lack of *Tas1r3*/*Sac* polymorphic effects on the preference of high concentrations of sucrose probably is not related to taste perception, because impulse discharge of the chorda tympani nerve to lingual application of sweeteners progressively increased across the whole range of tested concentrations of stimuli [30]. Potentially, the influence of *Tas1r3* polymorphisms can be masked by responses to the calorie content of test solutions, which does not depend on T1R2/T1R3-mediated perception, such as enhancement of

the cephalic phase of insulin release [45], or by conditional reflexes to nongustatory stimuli, such as smell and texture [46]. Intake of nonnutritive sweeteners likely does not induce strong responses such as these. Although excitations induced by nutritive or nonnutritive sweeteners are relayed to the brain through common primary pathways, they interact with different brain regions [47].

In the experimental model used here, an influence of *Tas1r3* hemizygosity should also be considered. Hemizygosity can lead to haplo-insufficiency, wherein a single functional copy of a gene is insufficient to maintain the standard phenotype [48]. In this connection, we estimated the effect of absence of a single *Tas1r3* allele. In the BALT, the impact of the *Tas1r3* gene hemizygosity was not detected, as we showed comparing responses between the B6 strain and B6-*Tas1r3*+/- hybrids (Fig 1*E*–1*G*). Moderate signs of *Tas1r3* haplo-insufficiency were found in the long-term 2-BT. Lack of the T allele in B6-*Tas1r3*+/- mice was associated with reduced consumption of intermediate (4–8%) and preference for lower concentrations of sucrose (1–2%) and saccharin (0.2–2 mmol/L) relative to the B6 strain. However, the difference between *Tas1r3* homozygous and hemizygous mice was notably smaller than that between 129S2B6F1 and 129S2B6-*Tas1r3*-/-F1 hybrids (Fig 2).

The central finding of our study showed that the dominant *Tas1r3* allele reduces plasma glucose concentration after enteral or parenteral glucose administration. Conversely, 129S2B6-*Tas1r3*-/-F1 hybrids, carrying the recessive allele, exhibited lower glucose tolerance (Fig 4*A and* 4*B*). To distinguish between effects of *Tas1r3* functional polymorphisms and haplo-insufficiency, we additionally compared responses of the B6 parental strain (*Tas1r3* homozygote) and B6-*Tas1r3*+/- hemizygote and found no difference between the groups in the IP GTT (Fig 4*C*). Surprisingly, after IG load with glucose, B6-*Tas1r3*+/- hemizygotes exhibited lower plasma glucose concentrations, which can reflect reduced expression of SGLT1 because of decreased GLP1 and GIP production in the small intestine mucosa [6, 49] rather than a consequence of impaired utilization of glucose in tissues. Altogether, the analyses we performed suggest that the reduction of glucose tolerance in 129S2B6-*Tas1r3*-/-F1 hybrids is due to expression of the recessive NT allele. At the same time, *Tas1r3* genotype did not influence baseline glucose level in nonfasting mice (Fig 5*A*). As we reported earlier, insulin sensitivity was reduced in the *Tas1r3*-knockout mice [16]; however, in the present assay, there was no relationship between tissue sensitivity to insulin and *Tas1r3* genotype (Fig 4*E and* 4*F*).

Potentiation of insulin secretion in islet cells of the pancreas and incretin secretion in the small intestine, dependent on the T1R3 receptor protein, are supposed to be important mechanisms of the influence of postoral sweet taste receptors on glucose tolerance [26, 50, 51]. In the present study, plasma insulin level was assessed in F1 crosses between 129P and B6By substrains, which did not differ in glucose tolerance from 129S2 and B6 substrains, respectively (Fig 3). Plasma insulin level declined in 129PB6-*Tas1r3*-/-F1 hybrids carrying the recessive allele relative to 129PB6ByF1 mice (Fig 6*A*), which corresponds to the attenuation of glucose tolerance in 129S2B6-*Tas1r3*-/-F1 hybrids. However, reduction of insulin level was due to haplo-insufficiency, as confirmed by comparison between the B6 strain and B6-*Tas1r3*+/- mice (Fig 6*D*).

Ablation of the T1R3 receptor affects anabolism, including lipid biosynthesis. Although deletion of the T1R2 or T1R3 protein did not influence body weight in B6 mice maintained on the normocaloric diet [52, 53], there was an increase in epididymal fat weight and signs of dyslipidemia, including elevation of plasma triglycerides and increased glycerol levels [53]. In contrast, when mice were fed the high-calorie diet, impaired sweet taste perception slowed down body and adipose tissue weight gain [54, 55]. This result is consistent with data obtained in experiments with adipocyte cell cultures, which have confirmed that the T1R3 protein mediates a stimulative influence of artificial sweeteners on adipogenesis [51, 56]. Comparison

between 129S2B6F1 and 129S2B6-*Tas1r3*-/-F1 hybrids performed in this study did not reveal a relation between *Tas1r3* polymorphisms and body, fat, and liver mass (Fig 5B–5D) or triglyceride levels (Fig 6B). However, *Tas1r3* hemizygosity caused an increase of body, fat, and liver mass (Fig 5F–5H), allowing us to speculate that, in the experimental model used here, the recessive *Tas1r3* allele confers reduction of body, liver, and fat mass, which is masked by haplo-insufficiency.

## Concluding remarks

On the whole, the obtained data provide new *in vivo* evidence that polymorphisms of the *Tas1r3* gene (*Sac* locus) influence carbohydrate metabolism, as well as sweetener intake and preference. In particular, presence of the dominant *Tas1r3* allele ($Sac^b$ locus) on the 129S2B6F1 background genotype results in higher glucose tolerance, along with increased consumption and preference for sugars and artificial sweeteners during initial licking. Hemizygosity of *Tas1r3* does not affect these behavioral and metabolic traits. However, the loss of one functional *Tas1r3* allele was linked to decline in intake and preference for sweeteners in the long-term 2-BT, as well as reduction of plasma insulin, body mass, liver mass, and fat mass. Our results suggest that further investigation of postoral sweet detection with the T1R3 protein may lead to new understanding of carbohydrate homeostasis disorders.

## Acknowledgments

We thank Mrs. Irina E. Bogatyrova, D.V.M., for maintaining of the mouse colony.

## Author Contributions

**Conceptualization:** Vasiliy A. Zolotarev.

**Formal analysis:** Vladimir O. Murovets, Ekaterina A. Lukina, Raisa P. Khropycheva.

**Investigation:** Vladimir O. Murovets, Ekaterina A. Lukina, Egor A. Sozontov, Julia V. Andreeva, Raisa P. Khropycheva, Vasiliy A. Zolotarev.

**Methodology:** Vladimir O. Murovets, Ekaterina A. Lukina, Julia V. Andreeva, Vasiliy A. Zolotarev.

**Resources:** Julia V. Andreeva, Raisa P. Khropycheva.

**Software:** Egor A. Sozontov.

**Supervision:** Vasiliy A. Zolotarev.

**Validation:** Vladimir O. Murovets.

**Visualization:** Egor A. Sozontov.

**Writing – original draft:** Vasiliy A. Zolotarev.

**Writing – review & editing:** Vladimir O. Murovets, Vasiliy A. Zolotarev.

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
