## [Decision Letter · Decision Letter 0]

17 Mar 2020

PONE-D-20-05222

Allelic variation of the Tas1r3 taste receptor gene affects sweet taste responsiveness and metabolism of glucose in F1 mouse hybrids

PLOS ONE

Dear Dr. Vasiliy Zolotarev,

Thank you for submitting your manuscript to PLOS ONE. After careful consideration, we feel that it has merit but does not fully meet PLOS ONE’s publication criteria as it currently stands. Therefore, we invite you to submit a revised version of the manuscript that addresses the points raised during the review process.

As the reviewer suggests, basic information is needed regarding the sugar metabolism of the mouse F1 hybrids, strain of 129SV, B6, B6-Tas1r3(-/-).  The paper should totally be rewritten.

We would appreciate receiving your revised manuscript by May 01 2020 11:59PM. To enhance the reproducibility of your results, we recommend that if applicable you deposit your laboratory protocols in protocols.io, where a protocol can be assigned its own identifier (DOI) such that it can be cited independently in the future. For instructions see: http://journals.plos.org/plosone/s/submission-guidelines#loc-laboratory-protocols

We look forward to receiving your revised manuscript.

Kind regards,

Keiko Abe, Ph.D.

Academic Editor

PLOS ONE

Journal Requirements:

2.Thank you for stating the following in the Financial Support section of your manuscript:

"Supported by the Russian Foundation for Basic Research, grant 19-015-00121."

Please remove any funding-related text from the manuscript and let us know how you would like to update your Funding Statement. Currently, your Funding Statement reads as follows: "No"

Reviewers' comments:

Reviewer's Responses to Questions

**Comments to the Author**

1. Is the manuscript technically sound, and do the data support the conclusions?

Reviewer #1: No

Reviewer #2: Partly

2. Has the statistical analysis been performed appropriately and rigorously? 

Reviewer #1: No

Reviewer #2: Yes

3. Have the authors made all data underlying the findings in their manuscript fully available?

Reviewer #1: Yes

Reviewer #2: Yes

4. Is the manuscript presented in an intelligible fashion and written in standard English?

Reviewer #1: No

Reviewer #2: Yes

5. Review Comments to the Author

Reviewer #1: In this manuscript, authors appeared the difference of sweet taste preference and sugar metabolism among the crossed mouse strains: F1 male hybrids obtained from 129SvPasCrl, B6 and/or B6-Tas1r3(-/-). This manuscript lacks the basic information about the sugar metabolism about 129Sv, B6, B6-Tas1r3(-/-). Moreover, there is a major problem with the description of discussion. Therefore, I recommend that this manuscript should be rewritten entirely.

1. Authors should appear the difference about the sugar metabolism between 129SvPasCrl and B6. The data of 129S, 129SB6F1, 129SB6-Tas1r3-/-F1, B6, B6-Tas1r3+/-, and B6-Tas1r3-/-should be shown in one graph.

2. Discussion contains a lot of results. I think it is a place to discuss the cause of the difference among 129SB6F1, 129SB6-Tas1r3-/-F1, and B6-Tas1r3+/-.

Reviewer #2: In the present report, the authors used the mouse F1 hybrids between 129S2 and B6, or between 129S and B6-Tas1r3-KO strains, and examined the effects of the allelic variation of the Tas1r3 gene on the sweet taste perception and metabolism of glucose. I have a few comments on the manuscript which should be addressed to strength their conclusions.

1. Introduction on p4: In 35th line, I think the B6 strain is “Sacb homozygote” not “Sacd homozygote”.

2. The authors used the hybrids between B6By and 129 mice for Fig. 5a-c. The hybrid between 129S2 and B6 should be used here.

3. T1R3 is also the important component of the umami taste receptor (T1R1+T1R3). The authors need to consider the effect on the umami taste receptor especially when they discuss the postoral events.

6. PLOS authors have the option to publish the peer review history of their article (what does this mean?). If published, this will include your full peer review and any attached files.

Reviewer #1: No

Reviewer #2: No

---

## [Author Response · Author response to Decision Letter 0]

25 Apr 2020

PONE-D-20-05222

Revision 1

Responses to Reviewers #1 and #2

Response to Reviewer #1.

Reviewer #1: In this manuscript, authors appeared the difference of sweet taste preference and sugar metabolism among the crossed mouse strains: F1 male hybrids obtained from 129SvPasCrl, B6 and/or B6-Tas1r3(-/-). This manuscript lacks the basic information about the sugar metabolism about 129Sv, B6, B6-Tas1r3(-/-). Moreover, there is a major problem with the description of discussion. Therefore, I recommend that this manuscript should be rewritten entirely.

1. Authors should appear the difference about the sugar metabolism between 129SvPasCrl and B6. The data of 129S, 129SB6F1, 129SB6-Tas1r3-/-F1, B6, B6-Tas1r3+/-, and B6-Tas1r3-/-should be shown in one graph.

 Response: 

Reviewer #1 insists on additional information but does not dispute the relevance of the main goal and provides no critique of experimental results, their evaluation, and our conclusions. We also consider the recommendation of Reviewer #1 “that this manuscript should be rewritten entirely” to be excessive.

The main goal of the current study was to define whether variations in a single taste receptor gene (Tas1r3) play a role in the regulation of glucose tolerance, insulin resistance, and some other parameters of metabolism. Developing the experimental model, we considered that the maintenance of glucose homeostasis involves a complex interplay of many genes and their actions in response to exogenous stimuli. Thus, the use of inbred mouse strains seems nonrelevant for definition of the role of the selected gene under physiological conditions. That is why the current study is based on comparisons between hybrids bearing different sets of Tas1r3 dominant and recessive alleles but having identical background genotypes. We believe that the constant background genotype of hybrid F1 groups is an indispensable condition to rigorously characterize effects of Tas1r3 polymorphisms on glucose homeostasis. 

In this paradigm, “basic information about the sugar metabolism” (maybe glucose metabolism?) of the parental strains B6, B6-Tas1r3-/-, and 129 seems unnecessary to draw correct conclusions. Due to the polygenic nature of regulation of glucose homeostasis, parental strains would not be appropriate controls for polymorphic effects of the Tas1r3 gene. 

Furthermore, differences in glucose homeostasis (including glucose and insulin tolerance) between B6 and 129 inbred mouse strains and substrains are well characterized elsewhere (e.g., Almind and Kahn, 2004; Berglund et al., 2008; Toye et al., 2005; Goren et al., 2004; Kadota et al., 2016). One may also find the information in the Mouse Phenome Database of Jackson Laboratory (phenome.jax.org/projects/Eumorphia3/strains). In the literature, there are several examples of the effect of Tas1r3 deletion on glucose metabolism regulation (e.g., Simon et al., 2014; Murovets et al., 2015). 

At the same time, addressing comment 1 of Reviewer #1, we have added a supplementary graph (S1_Fig.) demonstrating glucose tolerance in C57BL/6J, C57BL/6ByJ, 129P3/J, 129Sv, and knockout B6-Tas1r3-/- mouse strains. The graph shows significant differences in responses between 129 and B6 inbred strains and overlap within substrains. Additionally, S1Fig. shows a significant impact of Tas1r3 gene deletion. 

Moreover, to address comment 1 and to make the paper clearer to nonspecialists, we have revised the Introduction section, by adding the following at line 119 of the Revised Manuscript with tracked changes: “Studies have shown that plasma glucose level was considerably higher in the B6 strain than in 129 (Almind and Kahn, 2004; Berglund et al., 2008; phenome.jax.org/ projects/Eumorphia3/strains). Mice from the B6 strain show mild glucose intolerance and impaired insulin secretion despite normal insulin sensitivity (Toye et al., 2005). In contrast, the 129 substrains exhibited higher glucose tolerance (Almind and Kahn, 2004; phenome.jax.org/projects/Eumorphia 3strains). Likewise, the B6 mice were more insulin resistant during the insulin tolerance test, whereas 129SVE mice were relatively insulin sensitive (Almind and Kahn, 2004)”.

To emphasize importance of the influences of the animals’ genetic background on the impact of genetic manipulation with a particular gene (Sittig et al., 2016), we also edited the conclusions by specifying the background genotype: at line 421: “In particular, presence of the dominant Tas1r3 allele (Sacb locus) on the 129S2B6F1 background genotype results in higher glucose tolerance, along with increased consumption and preference for sugars and artificial sweeteners during initial licking”.

Reviewer #1 wrote: “2. Discussion contains a lot of results. I think it is a place to discuss the cause of the difference among 129SB6F1, 129SB6-Tas1r3-/-F1, and B6-Tas1r3+/-.”

Response:

The guidelines for authors for PLOS ONE do not give specific recommendations on how to prepare the Discussion section. That is why we followed the Springer Nature author and reviewer tutorials (available at https://www.springer.com/gp/authors-editors/authorandreviewertutorials/writing-a-journal-manuscript/discussion-and-conclusions/10285528). Following this guide, we did not duplicate raw data but discussed their genetic nature in connection with taste receptor gene Tas1r3 allelic variations or effects of Tas1r3 haplo-insufficiency. In the Discussion we also compared our results and conclusions with those from other investigations in this field and analyzed limitations of the study. 

To address “the cause of the difference” between the assessed genotypes, we substantially expanded the Introduction section to include the following starting at line 50 of the marked-up manuscript: 

“There is good evidence that the sweet taste receptor proteins, along with several signal-transducing molecules, are expressed not only in taste buds on the tongue but also in organs regulating metabolism (Laffitte et al. 2014). Sweet taste receptor subunits T1R2 and T1R3 are coexpressed in enteroendocrine cells of the small intestine (Jang et al., 2007; Margolskee et al., 2007). Their activation by natural sugars and artificial sweeteners leads to secretion of gut hormones such as glucagon-like peptide (GLP-1), GLP-2, and the gastric inhibitory peptide, acting as incretins to enhance insulin secretion (Jang et al., 2007; Kokrashvili et al., 2009). In the intestine, T1R3 is also involved in regulation of expression of sodium-dependent glucose transporter isoform 1 (Margolskee et al., 2007). Coupling of T1R3 with T1R1 produces a heterodimer that serves as an intestinal L-amino acid sensor modulating cholecystokinin release (Daly et al., 2013).

The T1R3 protein is detected in mouse pancreatic β-cells, where its expression level is greater than T1R2, suggesting that most likely T1R3 is present in homodimeric form or is more sensitive to glucose heteromers when coupled with a calcium-sensing receptor (Kyriazis et al. 2012; Maitrepierre et al. 2012; Damak et al. 2003). In pancreatic β-cells, stimulation of the sweet taste receptor elicits insulin release by elevating intracellular Ca2+ and/or cAMP (Kojima and Nakagawa, 2011) synergetic to the major universal mechanism of metabolic detection of glucose, which includes the glucose transporter GLUT2, glucokinase, and the ATP-sensitive K+ (KATP) channel (Craig et al. 2008). In β-cells, T1R3 can be also dimerized with the T1R1 subunit, forming the receptor detecting amino acids, which stimulates insulin synthesis and reduces cell apoptosis, affecting the mechanistic target of rapamycin complex 1 (mTORC1) through activation of phospholipase C and extracellular signal-regulated protein kinase 1 (ERK1) and ERK2 (Wauson et al., 2012). Importantly, the knockout of T1R3 in mice resulted in substantial impairment of metabolism and cell proliferation. The Tas1r3-/- deletion decreased glucose tolerance (Simon et al., 2014; Murovets et al. 2015) and increased insulin resistance (Murovets et al. 2014). T1R3 knockout animals are resistant to sucrose-induced obesity and have smaller fat depots when fed a high-sucrose diet (Simon et al., 2014). Deletion of Tas1r3-/- reduces activation of mTORC1 (Wauson et al., 2012) and decreases density of pancreatic islets and expression of caspase 3 (Murovets et al., 2019)”.

Responding to question 2, Reviewer #1 evaluated the performed statistical analysis as inappropriate and non-rigorous. 

We regret that the Reviewer gave no examples to support this statement. In performing the analysis of data from GTTs and ITTs we addressed the guidelines for mouse metabolic phenotyping in diabetes research, which specify that changes in glucose levels over time during a GTT or ITT are usually analyzed using two-way ANOVA, and AUCs are presented for GTT to draw reliable conclusions (Alquier and Poitout 2018). That is why we applied the ANOVA test in the current research, like we did in our similar paper published in PLOS ONE (Murovets et al., 2015). 

Question 4

We are surprised at the negative answer of the Reviewer #1 to question 4, regarding quality of standard English. Of course, there are no native English speakers among the authors. That is why our papers are always checked by a native English speaker with appropriate background and experience, and the current manuscript is no exception. With Revision 1, we attach a supplementary file with a letter attesting to appropriate editing for the initial submission and the revision 1 (S2_Murovets_proofreading).

We believe that our answers to the comments will facilitate further impartial evaluation of the manuscript.

Vasiliy A. Zolotarev, Ph.D.

Selected references

Almind K, Kahn CR. Genetic determinants of energy expenditure and insulin resistance in diet-induced obesity in mice. Diabetes. 2004 Dec;53(12):3274-85.

Alquier T, Poitout V. Considerations and guidelines for mouse metabolic phenotyping in diabetes research. Diabetologia. 2018 Mar;61(3):526-538. doi: 10.1007/s00125-017-4495-9.

Berglund ED, Li CY, Poffenberger G, Ayala JE, Fueger PT, Willis SE, Jewell MM, Powers AC, Wasserman DH. Glucose metabolism in vivo in four commonly used inbred mouse strains. Diabetes. 2008 Jul;57(7):1790-9.

Goren HJ, Kulkarni RN, Kahn CR. Glucose homeostasis and tissue transcript content of insulin signaling intermediates in four inbred strains of mice: C57BL/6, C57BLKS/6, DBA/2, and 129X1. Endocrinology. 2004 Jul;145(7):3307-23. 

Kadota Y, Kawakami T, Takasaki S, Sato M, Suzuki S. Gene expression related to lipid and glucose metabolism in white adipose tissue. Obes Res Clin Pract. 2016 Jan-Feb;10(1):85-93. doi: 10.1016/j.orcp.2015.04.009.

Sittig LJ, Carbonetto P, Engel KA, Krauss KS, Barrios-Camacho CM, Palmer AA. Genetic background limits generalizability of genotype-phenotype relationships. Neuron. 2016; 91:1253–1259.

Toye AA, Lippiat JD, Proks P, Shimomura K, Bentley L, Hugill A, Mijat V, Goldsworthy M, Moir L, Haynes A, Quarterman J, Freeman HC, Ashcroft FM, Cox RD. A genetic and physiological study of impaired glucose homeostasis control in C57BL/6J mice. Diabetologia 2005, 48(4), 675–686.

 

Response to Reviewer #2.

Reviewer #2: In the present report, the authors used the mouse F1 hybrids between 129S2 and B6, or between 129S and B6-Tas1r3-KO strains, and examined the effects of the allelic variation of the Tas1r3 gene on the sweet taste perception and metabolism of glucose. I have a few comments on the manuscript which should be addressed to strength their conclusions.

1. Introduction on p4: In 35th line, I think the B6 strain is “Sacb homozygote” not “Sacd homozygote”.

Response: Corrected, thank you.

2. The authors used the hybrids between B6By and 129 mice for Fig. 5a-c. The hybrid between 129S2 and B6 should be used here.

Response: The goal of the study was to elucidate whether polymorphisms of the single taste receptor gene affect carbohydrate metabolism. 

The maintenance of glucose homeostasis involves a complex interplay of many genes and their actions in response to exogenous stimuli. To exclude polygenic effect on the phenotype and focus on effects of allelic variations of the single gene, we chose a hybrid F1 model with different sets of Tas1r3 dominant and recessive alleles or that lacked the Tas1r3 gene but had identical background genotypes.

Parental B6 substrains (C57BL/6J (B6) and C57BL/6ByJ (B6By)) carry the same Tas1r3 dominant gene, while 129 substrains (129SvPasCrl (129S2) and 129P3/J (129P)) are Tas1r3 recessive homozygotes. For this reason we consider it acceptable to use different parental substrains to illustrate the possibility of polymorphic effects. Please note that, in analyzing a particular trait, we used hybrids obtained from the same substrains, fully matching in background genotype. 

On the other hand, an animal’s genetic background influences the impact of genetic manipulation with a particular gene. That is why, additionally, we compared glucose tolerance between parental substrains. Supplementary graph S1Fig. demonstrates that there are no differences in IP glucose tolerance between 129P and 129S2 substrains, or between B6 and B6By.

Line 297 of the Revised Manuscript with tracked changes now has the following addition: 

“Additional analysis of IP glucose tolerance was performed in parental strains: 129P, 129S2, B6, B6By, and B6-Tas1r3-/-. In the nonfasting state, parental strains differed in baseline glucose level, which was higher in the B6 strain, including the knockout substrain, than in 129 (S1Fig.). However, there was significant overlap between responses of B6 and B6By, as well as between 129P and 129S2. The IP glucose tolerance was substantially reduced in the B6 strain compared to 129, but no significant differences were found within strains excluding B6-Tas1r3-/-. The Tas1r3 gene deficiency markedly worsens glucose tolerance compared both to B6 and to B6By (S1Fig). Comparison of Fig. 3a and S1Fig. demonstrates that hybrid groups had intermediate glucose tolerance between parental strains.”

3. T1R3 is also the important component of the umami taste receptor (T1R1+T1R3). The authors need to consider the effect on the umami taste receptor especially when they discuss the postoral events.

Response:

In the manuscript, we do not address mechanisms of metabolic control involving the T1R3 receptor protein. However, we fully agree with the Reviewer #2 that coupling of T1R3 with T1R1 in the small intestine or pancreas can specifically activate intracellular signaling cascades regulating hormone secretion and cell proliferation. To emphasize it, we included the following references in the revised manuscript.

Line 58 of the marked-up manuscript: “Coupling of T1R3 with T1R1 produces heterodimer that serves as an intestinal L-amino acid sensor modulating cholecystokinin release (Daly et al., 2013)”.

Line 67 of the marked-up manuscript: “In β-cells, T1R3 can be also dimerized with the T1R1 subunit, forming the receptor detecting amino acids, which stimulates insulin synthesis and reduces cell apoptosis affecting the mechanistic target of rapamycin complex 1 (mTORC1) through activation of phospholipase C and extracellular signal-regulated protein kinase 1 (ERK1) and ERK2 (Wauson et al., 2012). ... Deletion of Tas1r3-/- reduces activation of mTORC1 (Wauson et al., 2012) and decreases density of pancreatic islets and expression of caspase 3 (Murovets et al., 2019)”.

Sincerely,

Vasiliy A. Zolotarev, Ph.D.

---

## [Decision Letter · Decision Letter 1]

26 May 2020

PONE-D-20-05222R1

Allelic variation of the Tas1r3 taste receptor gene affects sweet taste responsiveness and metabolism of glucose in F1 mouse hybrids

PLOS ONE

Dear Dr. Vasiliy Zolotarev,

Thank you for submitting your manuscript to PLOS ONE. After careful consideration, we feel that it has merit but does not fully meet PLOS ONE’s publication criteria as it currently stands. Therefore, we invite you to submit a revised version of the manuscript that addresses the points raised during the review process.

One of the reviewers has evaluated the revised manuscript, but further revision will be needed at the following points, she/he says."discussion about the validity of comparison between the data from 129B6 vs 129B6-T1r3KO and the data from B6 vs B6-T1r3KO  are still insufficient"Considering these comments, the authors are expected to prepare a revised manuscript one again.

We look forward to receiving your revised manuscript.

Kind regards,

Keiko Abe, Ph.D.

Academic Editor

PLOS ONE

Reviewers' comments:

Reviewer's Responses to Questions

**Comments to the Author**

1. If the authors have adequately addressed your comments raised in a previous round of review and you feel that this manuscript is now acceptable for publication, you may indicate that here to bypass the “Comments to the Author” section, enter your conflict of interest statement in the “Confidential to Editor” section, and submit your "Accept" recommendation.

Reviewer #1: All comments have been addressed

Reviewer #2: (No Response)

2. Is the manuscript technically sound, and do the data support the conclusions?

Reviewer #1: Partly

Reviewer #2: Yes

3. Has the statistical analysis been performed appropriately and rigorously? 

Reviewer #1: Yes

Reviewer #2: Yes

4. Have the authors made all data underlying the findings in their manuscript fully available?

Reviewer #1: Yes

Reviewer #2: Yes

5. Is the manuscript presented in an intelligible fashion and written in standard English?

Reviewer #1: Yes

Reviewer #2: Yes

6. Review Comments to the Author

Reviewer #1: Thank you for the explanation. I understand authors’ purpose to clarify the metabolic effects of T1r3 by polymorphisms by comparing 129B6 and 129B6/129B6-T1r3KO.

However, I still wonder if the structure of this manuscript should be as it is.

1. I was able to understand that the glucose tolerance of 129 and B6 is completely different since authors provided Figure S1. I would like Figure S1 to be included in the main text for general readers of PLOS ONE. Is it correct to assume that the glucose metabolism of 129B6 and B6 is also different? I recommend that authors discuss about the validity of comparison between the data from 129B6 and 129B6-T1r3KO and the data form B6 and B6/B6-T1r3KO, even though the physiological background of 129B6 and B6/B6 are not equivalent.

2. As authors mentioned, the results are not written twice. However, I think that some sentences should be written in results (L.337-352, L404-410). In addition, more discussion of the function of the polymorphisms in T1r3 should be needed. As for the additional information in introduction (L.50-L.76), I think it is necessary to discuss how these molecules are affected by polymorphisms in T1r3. I would like authors to explain the sentences of L.383 to L.387 to correspond the introduction (L.50-L.76).

Reviewer #2: (No Response)

7. PLOS authors have the option to publish the peer review history of their article (what does this mean?). If published, this will include your full peer review and any attached files.

Reviewer #1: No

Reviewer #2: No

---

## [Author Response · Author response to Decision Letter 1]

15 Jun 2020

PONE-D-20-05222R1

Revision 2

Responses to Reviewers 

Response to Reviewer #1.

Reviewer #1 wrote: “I was able to understand that the glucose tolerance of 129 and B6 is completely different since authors provided Figure S1. I would like Figure S1 to be included in the main text for general readers of PLOS ONE”. 

Response: We agree with the reviewer that addition of figure demonstrating glucose tolerance in parent strains could be useful for general readers. For this purpose, we inserted Fig. 3 and edited Table 2 (L. 295 of the Revised manuscript with tracked changes): “Initial analysis of IP glucose tolerance was performed in parental strains: 129P, 129S2, B6, B6By, and B6-Tas1r3-/-. In the nonfasting state, parental strains differed in baseline glucose level, which was higher in the B6 strain, including the knockout substrain, than in 129 (Fig. 3; p<0,05). However, there was significant overlap between responses of B6 and B6By, as well as between 129P and 129S2. The IP glucose tolerance was substantially reduced in the B6 strain compared to 129 (p<0.005), but no significant differences were found within strains excluding B6-Tas1r3-/-. The Tas1r3 gene deficiency markedly worsens glucose tolerance (Fig. 3) compared both to B6 and to B6By (p<0.01).

Comparison of Fig. 3 and Fig. 4a demonstrates that hybrid groups had intermediate glucose tolerance between parental strains.” 

The Reviewer #1:” Is it correct to assume that the glucose metabolism of 129B6 and B6 is also different? I recommend that authors discuss about the validity of comparison between the data from 129B6 and 129B6-T1r3KO and the data form B6 and B6/B6-T1r3KO, even though the physiological background of 129B6 and B6/B6 are not equivalent”.

Response: Yes, it is correct. For this reason, we did not compare 129B6-Tas1r3KO with B6-T1r3+/-. Distinctions between 129B6 and 129B6-Tas1r3KO hybrids were analyzed to assess effects of dominant and recessive alleles of the Tas1r3 gene. B6 and B6-Tas1r3+/- mice differed in number of dominant Tas1r3 genes. Using B6 and B6-Tas1r3+/- groups, we primarily evaluated the effect of the Tas1r3 gene dosage (hemizygosity). In the available literature, there are no data on interaction between Tas1r3 and other genes that supports validity of comparison between groups with identical B6 genetic background and, to some extent, allows to transfer results to 129B6 hybrids. 

Reviewer #1: “As authors mentioned, the results are not written twice. However, I think that some sentences should be written in results (L.337-352, L404-410)”. 

Response: In the manuscript, we tried to keep the following scheme: the Results section describes responses of hybrids, whereas in the Discussion section we analyze effects of the Tas1r3 genotype and compare the obtained data with literature. That is why we would like to leave mentioned phrases in place. 

Reviewer #1: In addition, more discussion of the function of the polymorphisms in T1r3 should be needed. As for the additional information in introduction (L.50-L.76), I think it is necessary to discuss how these molecules are affected by polymorphisms in T1r3.

Response: To our knowledge, almost all publications discussing effects of the mouse gene Tas1r3 polymorphisms address to the detailed analysis performed by Danielle Reed et al. (2004), which we cited in the current manuscript. We edited new revision by adding the following (L. 95): “Allelic variants of the Tas1r3 gene correspond mainly to three nonsynonymous single nucleotide polymorphisms (SNPs) which do not act by blocking gene expression, changing alternative splicing, or interfering with protein translation in taste tissue. Among the polymorphisms, T179C, which causes a substitution of isoleucine to threonine at position 60 in the extracellular N domain of the T1R3 protein, influence the ability of the protein to form dimers or bind sweeteners (Reed et al. 2004)”. Further we wrote (L. 100): “This substitution reduces in vitro binding of T1R3 to caloric or noncaloric sweeteners, increasing, for instance, effective dose for sucrose up to 1000% (Nie et al. 2005).” 

Reviewer #1: I would like authors to explain the sentences of L.383 to L.387 to correspond the introduction (L.50-L.76). 

Response: When glucose is applied intragastrically in the GTT, sweet taste receptors in the small intestine are involved in reinforcement of secretion of gut hormones GLP-1 and GIP, which in turn facilitate insulin output in the pancreas and up-regulate expression of the intestinal glucose transporter SGLT1. To clarify it, we edited the Introduction section (L. 57): “In the intestine, T1R3 upregulates through GLP-1 and GIP expression of sodium-dependent glucose transporter isoform 1 (SGLT1) important for the provision of glucose to the body and avoidance of intestinal malabsorption (Margolskee et al., 2007)”. When glucose is applied intraperitoneally, it affects sweet taste receptors in the pancreas stimulating insulin production but does not interact with intestinal receptors. We believe that impairment of T1R3-mediated sensitivity in B6-Tasr3+/- can both elevate plasma glucose level due to reduction of insulin output and suppress glucose absorptive capacity of the intestine. Thus, in hemizygotes after intraperitoneal load with glucose, reduced intestinal absorption can be discussed as a reason of lower plasma glucose level found in our experiments (L. 386): “B6-Tas1r3+/- hemizygotes exhibited lower plasma glucose concentrations, which can reflect reduced expression of SGLT1 because of decreased GLP1 and GIP production in the small intestine mucosa (Margolskee et al. 2007; Sigoillot et al. 2012), rather than a consequence of impaired utilization of glucose in tissues”. However, additional studies of this phenomenon are needed beyond the scope of this article.

We believe that translational value of our findings is that they show for the first time that allele variant of the Tas1r3 gene alter in vivo glucose metabolism in mice, and that similar relationships may exist in humans. 

Sincerely,

Vasiliy Zolotarev, Ph.D.

---

## [Decision Letter · Decision Letter 2]

25 Jun 2020

Allelic variation of the Tas1r3 taste receptor gene affects sweet taste responsiveness and metabolism of glucose in F1 mouse hybrids

PONE-D-20-05222R2

Dear Dr. Vasiliy Zolotarev,

We’re pleased to inform you that your manuscript has been judged scientifically suitable for publication and will be formally accepted for publication once it meets all outstanding technical requirements.

Kind regards,

Keiko Abe, Ph.D.

Academic Editor

PLOS ONE

Additional Editor Comments (optional):

Reviewers' comments:

Reviewer's Responses to Questions

**Comments to the Author**

1. If the authors have adequately addressed your comments raised in a previous round of review and you feel that this manuscript is now acceptable for publication, you may indicate that here to bypass the “Comments to the Author” section, enter your conflict of interest statement in the “Confidential to Editor” section, and submit your "Accept" recommendation.

Reviewer #1: All comments have been addressed

2. Is the manuscript technically sound, and do the data support the conclusions?

Reviewer #1: Yes

3. Has the statistical analysis been performed appropriately and rigorously? 

Reviewer #1: Yes

4. Have the authors made all data underlying the findings in their manuscript fully available?

Reviewer #1: Yes

5. Is the manuscript presented in an intelligible fashion and written in standard English?

Reviewer #1: Yes

6. Review Comments to the Author

Reviewer #1: (No Response)

7. PLOS authors have the option to publish the peer review history of their article (what does this mean?). If published, this will include your full peer review and any attached files.

Reviewer #1: No

---

## [Editor Report · Acceptance letter]

6 Jul 2020

PONE-D-20-05222R2 

Allelic variation of the Tas1r3 taste receptor gene affects sweet taste responsiveness and metabolism of glucose in F1 mouse hybrids 

Dear Dr. Zolotarev:

I'm pleased to inform you that your manuscript has been deemed suitable for publication in PLOS ONE. Congratulations! Your manuscript is now with our production department. 

Kind regards, 

on behalf of

Prof. Keiko Abe 

Academic Editor

PLOS ONE